# *Trypanosoma cruzi* Sirtuin 2 as a Relevant Druggable Target: New Inhibitors Developed by Computer-Aided Drug Design

**DOI:** 10.3390/ph16030428

**Published:** 2023-03-10

**Authors:** Glaucio Monteiro Ferreira, Thales Kronenberger, Vinicius Gonçalves Maltarollo, Antti Poso, Fernando de Moura Gatti, Vitor Medeiros Almeida, Sandro Roberto Marana, Carla Duque Lopes, Daiane Yukie Tezuka, Sérgio de Albuquerque, Flavio da Silva Emery, Gustavo Henrique Goulart Trossini

**Affiliations:** 1Department of Pharmacy, School of Pharmaceutical Sciences, University of São Paulo, Av Prof Lineu Prestes 580, Building. 13, São Paulo 05508-000, SP, Brazil; gmf@usp.br (G.M.F.);; 2Department of Clinical and Toxicological Analyses, School of Pharmaceutical Sciences, University of São Paulo, Av Prof Lineu Prestes 580, Building. 17, São Paulo 05508-000, SP, Brazil; 3Department of Oncology and Pneumonology, Internal Medicine VIII, University Hospital Tübingen, Otfried-Müller-Straße 10, 72076 Tübingen, Germany; 4School of Pharmacy, Faculty of Health Sciences, University of Eastern Finland, 70211 Kuopio, Finland; 5Department of Pharmaceutical Products, Faculty of Pharmacy, Federal University of Minas Gerais, Av. Antônio Carlos 6627, Belo Horizonte 31270-901, MG, Brazil; 6Department of Biochemistry, Institute of Chemistry, University of São Paulo, Av Prof Lineu Prestes 748, Building 12, São Paulo 05508-000, SP, Brazil; vitorma3@gmail.com (V.M.A.);; 7Department of Clinical Toxicological and Bromatological Analysis, School of Pharmaceutical Sciences of Ribeirão Preto, University of São Paulo, Av. do Café, Ribeirão Preto 14040-903, SP, Brazil; 8Department of Pharmaceutical Sciences, School of Pharmaceutical Sciences of Ribeirão Preto, University of São Paulo, Av. do Café, Ribeirão Preto 14040-903, SP, Brazil

**Keywords:** anti-infectives, computer-aided drug discovery, Sirtuin 2, *Trypanosoma cruzi*

## Abstract

*Trypanosoma cruzi*, the etiological agent of Chagas disease, relies on finely coordinated epigenetic regulation during the transition between hosts. Herein we targeted the silent information regulator 2 (Sir2) enzyme, a NAD^+^-dependent class III histone deacetylase, to interfere with the parasites’ cell cycle. A combination of molecular modelling with on-target experimental validation was used to discover new inhibitors from commercially available compound libraries. We selected six inhibitors from the virtual screening, which were validated on the recombinant Sir2 enzyme. The most potent inhibitor (CDMS-01, IC_50_ = 40 μM) was chosen as a potential lead compound.

## 1. Introduction

Neglected tropical diseases (NTDs) are a diverse group of diseases that prevail in tropical and subtropical countries, affecting around 1 billion people worldwide. Among those, around 8 million cases are associated with the parasite *Trypanosoma cruzi*, which causes the Chagas disease [1,2]. NTDs are currently controlled through vector elimination, using insecticides, serological blood screening and social development of living conditions. In terms of chemotherapy, Chagas disease is limited to the use of nifurtimox and benznidazole, which are inefficient in the chronic phase, often present side effects and suffer from the lack of paediatric formulations [1,3,4]. Additionally, the expansion of *T. cruzi* resistant represent a serious public health problem. Therefore, there is an urgent need for new treatments and, therefore, the validation of new potential drug targets [5].

Trypanosomatids have a complex cell cycle in which they must respond to various environmental conditions, such as the different hosts and vectors. In this sense, transcription regulation, in particular the epigenetic machinery, is an important aspect of the parasite biology that can be used the development of new antiparasitic drugs [6,7]. Disrupting epigenetic dysregulation holds therapeutic promise for several diseases where precise regulation of the cell cycle and gene expression is crucial, including various types of cancer, obesity, and Parkinson’s disease [8,9,10]. Examples of epigenetic modifications include post-translational histone modifications, such methylation or acetylation, that modulate the chromatin opening state and, as consequence, the gene expression. Particularly, the acetylation of histone’s lysine residues can neutralize their positive charge, responsible for the interaction with DNA, leading to the relaxation of the chromatin, which ultimately regulates the availability of the DNA to the transcription machinery [10]. 

Many highly conserved enzymes responsible for histone’s post-translational modifications appear to be absent in trypanosomatids, whereas unusual histone sequences and histone-modifying proteins can be found [11]. Among the most common histone modification processes in parasites, it is worth mentioning the acetylation of the C-terminal histone H2A, the alanine methylation in the N-terminal histones H2A, H2B and H4 and tri-methylation of the lysine residues 4 and 76 of histone H3 [12]. The sirtuin 2 protein (Sir2), also called regulatory information silencer 2, is an NAD^+^-dependent N-acetyl-lysine deacetylase enzyme, which acts as relevant epigenetic regulator. Sirtuins catalyse the cleavage of the glycosidic linkage between nicotinamide and ADP-ribose of NAD^+^ followed by transfer of the acetyl group from lysine to ADP-ribose, generating a deacetylated lysine, nicotinamide and 2’- and 3’-O-acetyl-ADP-ribose [13].

Gaspar et al., (2018) genetically validated the *T. cruzi* Sir2-related protein 1 (*Tc*Sir2rp1) as essential for the parasite viability [14]. Their group also demonstrated the characteristic NAD^+^-dependent deacetylase activity and further its inhibition by the classic sirtuin inhibitors: nicotinamide and bisnaphthalimidopropyl-spermidine (BNIP) [14,15]. BNIP-spermidine, however, performed poorly in the mouse model of Chagas disease [14].

Since new drugs are required for the trypanosomiasis treatment and sirtuin inhibitors were shown to block the parasite proliferation [16], this study focused on the use of computer aided-drug design (CADD) approaches to discover small molecules with Sirtuin 2 inhibitory activity and antiparasitic activity. Our process started by creating a model of the *Tc*Sir2 structure. We then screened a database of commercially available drug-like compounds using filters and chose the best options by testing them in the *Tc*Sir2 model’s NAD+ binding site. We checked the stability of the selected compounds through simulations. Finally, we tested the most promising compounds in cells, both for their ability to kill the parasite and for any harmful side effects and checked their ability to inhibit Sir2.

## 2. Results and Discussion

### 2.1. TcSir2 Structure Harbours a Conserved NAD^+^ Binding Site

*Tc*Sir2 was modelled using the human Sir2 (PDB: 3ZGV) structure as a template, which presents 46% similarity and conserved motifs such as zinc-finger domain (Cys150, Cys155, Cys174 e Cys177). Recently, the *Leishmania infantum* Sir2 structure (*Li*Sir2, PDB: 5OL0 [17]) was revealed (deposited on 2017-07-26 and released on 2018-02-28). *Li*Sir2 shares 58% similarity with *Tc*Sir2 and presents conserved motifs such as zinc-finger domain and catalytic domain (Appendix A). Additionally, a search for suitable Sirtuin 2 three-dimensional structures was carried out at the Protein Bank Data (PDB). Structures with resolution < 2.5 Å, R-value free <0.3, R-value work <0.3, without inhibitor and with the cofactor (NAD^+^) were selected. Besides, we also selected wild-type (WT) structures with complete loops (Appendix A). 

Nevertheless, the others organisms’ structures (Leishmania amazonensis, Armadillidium nasatum, Bactrocera latifrons, Lygus Hesperus, Perkinsus olseni, Portunus trituberculatus, Rhizopus azygosporus, and Rhizopus stolonifer) offer new modelling opportunities to be explored in follow-up studies, which do not encompass the current approach. However, the Sir’s family present high similarity between its subtypes, like HDAC’s.

It is important to compare hSir2 and *Tc*Sir2 structures to identify elements promoting selectivity and avoid pan-sirtuin inhibitors. The sirtuin enzyme family members SIRT1, SIRT2, and SIRT3 have different expressions and play different roles in trypanosomatids, depending on the context and experimental conditions. Hence, it is not clear whether a selective inhibitor of sirtuin 2, which is more expressed in the amastigote form, will be better than non-selective ones for treating Chagas disease [18].

*Tc*Sir2 has a conserved binding site for the NAD^+^ cofactor region comprising three regions: (i) site A, where the adenine–ribose moiety of NAD^+^ binds; (ii) site B where the nicotinamide–ribose moiety binds; and (iii) site C where the nicotinamide is supposed to bind near to the active site of the enzyme [19,20]. Some differences in the binding sites were found by superimposing the comparative model of *Tc*Sir2, *Li*Sir2 and human Sir2 (hSir2) (Appendix A). ConSurf [21,22,23] analysis (Figure 1) reveals that the functional regions of this protein are highly conserved. For example, the residues which interact with NAD^+^, mainly in site C, in orange (Figure 1a), whereas hSir2 has 11 amino acids (Phe96, Leu103, Tyr104, Ile118, Phe119, Leu134, Asn160, Met168, His187 and Val266), and *Tc*Sir2 has 12 residues (Phe49, Arg50, Ile56, Pro68, Phe72, Val88, Asp92, Leu93, Asn123, Asp125, Gln217 and Val218). In this context, observing the complementarity of the cofactor with the residues for both hSir2 and *Tc*Sir2, it is noted that the trypanosomatids (*Tc*Sir2 and *Li*Sir2) showed more polar residues (Arg50, Asp92, Asn123, Asp125, and Gln217) than hSir2 (Asn160, Met168, and His187) and the same number of hydrophobic residues. MD analysis showed that Asp125, which is found in the trypanosomatids enzyme, would be relevant for the selectivity of Sir2 inhibitors.

The *Tc*Sir2 3D-model, including both the Zn^2+^ ion and the NAD^+^ cofactor, was submitted to molecular dynamics simulation, aiming to verify the stability of its secondary structure elements. MD simulations improved the model quality, by reducing the number of Ramachandran outliers (from 0.5% to none, Appendix A), and fixing the angles of Pro156 and Ser271, both in loop regions. The model was further validated and compared according to the quality indexes (Z-score −7.26 for *Tc*Sir2 before MD and −6.16 after MD) as determined by an Anolea server. Additionally, original disordered regions from the initial model folded into secondary structures upon simulation with explicit solvent (Figure 2a). C-terminal (residues 351–354) folded into another short strand complementing the existing (Tyr94–Gly96).

After that, *Tc*Sir2 and NAD^+^ analysis showed the modelled NAD^+^ cofactor kept hydrogen interactions with the Trp94 and Asp48 (Figure 2b,c), as well as ionic interactions between the Arg50′s sidechain (Site B) and double phosphate moieties. The free amide of the nicotinamide portion had a conserved water-mediated interaction with the Asp48 and with the Ile46′s backbone, while on the other side of the molecule, transient interactions between the sugar moiety bound to the adenine and Ser240 were observed. Avalos et al. (14) showed that π-cation and π-π interactions between Arg50 and adenine group and Phe49/Phe72 and the nicotinamide groups are relevant to stabilize the NAD^+^/Sir2 complex in the active conformation. MIF analyses corroborate this idea, presenting regions favorable for hydrophobic contacts observed at the nicotinamide binding site (Appendix A). H-bond acceptor-favorable regions (Appendix A) concentrate at Site B, which is occupied by the phosphate groups of NAD^+^.

Taking together this model with information about conserved residues (Appendix A), we suggest the relevant binding features cofactor NAD^+^ (Site A) and the catalytic site (Site C, nicotinamide site, Figure 3a). Altogether, the data supported the generation of two pharmacophore models, one based on whole NAD^+^-binding site (pharmacophore 1), and the second based on Site C (pharmacophore 2). Thus, two virtual screenings were performed to identify compounds capable of inhibiting *Tc*Sir2 (Figure 3b,c).

### 2.2. TcSir2 Inhibitor Virtual Screening and Experimental Validation

At this stage, purchasable and boutique subsets of the ZINC database [24] (35 million compounds) were employed for the virtual screening (Figure 3b). This was followed by molecular dynamics simulation, as a last selection step to exclude compounds with an unwanted/unstable binding mode, which did not respect the minimal interactions identified by the pharmacophore model. Four criteria were considered for the selection of possible inhibitors of *Tc*Sir2: (i) pharmacophore model complementarity; (ii) initial ranking by the docking score values; followed by (iii) visual inspection of the compound’s binding mode, analyzed according to interactions within the NAD^+^ binding site and site C; and (iv) stability within the active site along the MD simulation. MD simulations of *Tc*Sir2-CDMS-01 complexes (Figure 4a,b) showed conserved polar interactions with the Asp125 and Ser41, together with hydrophobic interactions with Phe98 and Ala38 near 3-fluoro-oxadiazole (Figure 4c). Those interactions were previously described as relevant for hSir2 binding with its substrate [25]. Additionally, the reduced RMSF values in the loop region between 122 to 144 (Figure 4d) of the *Tc*Sir2-CDMS-01 when compared with simulations of *Tc*Sir2-NAD^+^, suggests the loop stabilization upon ligand binding. Six compounds fitted these criteria (CDMS-05 and CDMS-06 (pharmacophore model 1—Figure 3c) and CDMS-01 to CDMS-04 (pharmacophore model 2—Figure 3d) and were selected for experimental validation.

Compounds were tested for *Tc*Sir2 inhibition by deacetylation, measured indirectly via fluorescent product formation (Appendix A and Table 1). Further, the kinetic inhibitory assays with the screening hits (CDMS-01 to CDMS-06) revealed that only compound CDMS-01 (IC_50_ = 40 μM, Figure 4e) efficiently inhibited *Tc*Sir2. Compounds CDMS-2 (IC_50_ = 240 μM), CDMS-3 (IC_50_ = 482 μM), and CDMS-6 (IC_50_ = 332.5 μM) had lower potency against *Tc*Sir2 under the conditions tested. In this context, CDMS-01 *K*_i_ (21.5 μM) is similar to the NAD^+^ *K*_m_ (25 μM), indicating that both would have the same affinity for the *Tc*Sir2 active site. Corroborating these results, a competitive inhibition mechanism for CDMS-01 was proposed (Appendix A), due to the constant V_max_ values and different K_m_ values throughout the experiments. This is supported by the complementarity of CDMS-01 with catalytic pocket CDMS-01 compound. All compounds were also tested on the ability to inhibit *T. cruzi* intracellular replication. Compound CDMS-01 inhibited recombinant *Tc*Sir2 (Figure 4e) and *T. cruzi* amastigote replication (Figure 4f) in a dose-response manner, with IC_50_ values of 40 and 76.7 µM, respectively. Furthermore, CDMS-01 inhibited the mammalian cell-lines with an IC_50_ value of 136 µM (Figure 4g), which indicates a slight selectivity (selectivity index = 1.77).

CDMS-01 is a new and promising prototype drug that targets a previously undruggable target. It provides a starting point for the development of new inhibitors with improved activity and selectivity.

From a safety point of view, further studies should be conducted in the future with other relevant cell lines. Assuming that the hit rate of a reported virtual screening and HTS studies is not applicable as a baseline for a random screening [19], the approach used in this work was able to discover new compounds with on-target inhibition capacity, membrane permeability, and activity in amastigotes form. However, these values are similar to the approved FDA drugs amphotericin B (SI_ama_ = 1.5) and meglumine antimoniate (SI_ama_ = 2.46) [26].

Surprisingly, compound CDMS-04 did not show activity against *Tc*Sir2, but it was active against amastigote with the highest selectivity index (IC_50_ = 15.16 µM and EC_50_ = 62.35 µM against LLCMK2 cells and SI = 4.11) compared to the other compounds from the screening. Hence, CDMS-04 is a candidate for optimization aiming at reducing toxicity increase selectivity. As an example, these studies developed a protocol in the search for potential hSir2 inhibitors that have bulky groups reported in their structure, showed inhibition for some sirtuin isoforms on the micromolar range (Sir1, Sir2, Sir3 and Sir5), but did not show inhibition in cellular assays [27,28,29]. However, it is possible that the CDMS compounds may act as PAN inhibitors, as the activity in the recombinant enzyme is not consistent with the effect on cells. Given our high IC_50_ values against *Tc*Sir2, we hypothesize that we could face multi-sirtuin inhibition, and further characterization would be needed to validate off-targets.

## 3. Materials and Methods

### 3.1. Computer-Aided Drug Discovery

#### 3.1.1. Comparative Modelling and Conservation Analysis

The sequence of the Sir2 from *Trypanosoma cruzi* CL Brener strain (herein referred to as *Tc*Sir2, accession code: XP_816094) was retrieved from the NCBI database and globally aligned against available protein structures using HHPred [30]. The human Sir2 crystal structure was selected as a template (PDB ID: 3ZGV, chain A, resolution 2.27 Å), due to having the highest similarity (46%) and the lowest number of outliers (0.3%). Sixty initial *Tc*Sir2 3D structure models were generated using the Modeller program [31], which were further refined by energy minimization with the steepest descent and rotamer variation. The models were ranked based on their calculated pseudo-energy values, and only the models with the lowest energy were further evaluated. The final model was chosen using Ramachandran plot analysis and Z-score validation and was tested for stability through molecular dynamics simulation. The evolutionary conservation of the *Tc*Sir2, hSir2, and *Li*Sir2 amino acid residues was calculated by comparing them to similar sequences using the ConSurf server [21,23]. It predicts conservation score of amino acid residues ranging from 1 to 9, which indicates the least to highest conserved, respectively (Figure 1).

#### 3.1.2. Molecular Dynamics Simulation

The selected model of *Tc*Sir2 underwent molecular dynamics (MD) simulations, using a previously described protocol [32]. *Tc*Sir2 was simulated in the presence and absence of the cofactor NAD^+^ to understand which amino acid residues contributed to the stability within the binding site (Figure 2). MD simulations were carried out using the Desmond engine [33] with the OPLS3e force field [34]. The simulated system encompassed the protein-ligand/cofactor complex, a predefined water model (TIP3P [35]) as a solvent and counter ions (Na^+^ or Cl^−^ adjusted to neutralize the overall system charge). The system was treated in an orthorhombic box with periodic boundary conditions specifying the shape and the size of the box as 13 Å distance from the box edges to any atom of the protein. We used a time step of 1 fs, the short-range coulombic interactions were treated using a cut-off value of 9.0 Å using the short-range method, while the smooth particle mesh Ewald method (PME) handled long-range coulombic interactions [36].

Root mean square deviation (RMSD) values for the protein backbone were monitored along the simulation, to infer simulation equilibration and protein changes (Appendix A). The trajectory was clustered using the protein backbone’s RMSD variance (cut-off 0.5 Å) resulting in one representative frame structure. In addition, the root mean squared fluctuation (RMSF) by residues was compared against the experimental B-factor and the observed fluctuation of residues like the reference structure that the models generated (Appendix A).

#### 3.1.3. Molecular Interactions Fields, Pharmacophore and Virtual Screening

MD’s representative conformation was used to analyze the distribution of molecular interaction fields (MIFs), performed using the GRID software (v22c, Molecular Discovery [37,38]). We evaluated three chemical probes (Dry, O_2_, and H_2_O) aiming to represent the most common intermolecular interactions (hydrophobic contacts, H-bond acceptor, and donor, respectively). Analyses were performed in a cubic box of 18 Å axis centered on the NAD^+^ cofactor coordinates. The generated box comprises the main residues that interact with NAD^+^ and Site C, ultimately aiming at the construction of two pharmacophoric models. MIF analyses were performed for both the human and trypanosomatid structures, and the trypanosomatid-specific regions were used to generate a pharmacophore model. Additionally, analysis of MD simulation trajectory suggests that not only hydrogen bonds, as MIFs originally proposed, but also π-cation and π-stacking interactions are involved in the NAD^+^ stabilization.

Subsequently, we generated two pharmacophore models, one based on the NAD+ binding site and another based on the substrate site. The NAD^+^ pharmacophore consisted of (i) three hydrogen bond acceptor points (red spheres, Figure 3d), (ii) a single hydrogen bond donor group point (blue sphere), and (iii) two hydrophobic/aromatic points (orange spheres), that can additionally represent the π-cation and π-stacking interactions. The other pharmacophore model encompassed the substrate catalytic site (Figure 3d) and considered: (i) a hydrogen bond acceptor group points (red sphere), (ii) two points of hydrogen bond donor groups (blue sphere), and (iii) three hydrophobic/aromatic points (yellow spheres) that can suggest π-cation and π-stacking interactions.

The virtual screening compound library was composed of all purchasable, all boutique subsets from the ZINC15 database [24] filtered by drug-like physicochemical properties. Namely, the number of hydrogen bond donors (HBD, with a range of 0 to 5) and acceptors (HBA, with a range of 0 to 10), the logarithm of the n-octanol/water partition coefficient (LogP, with a range of −2.5 to 5.5), and the permeability and net charge (between −2 and +2) of compounds predict their ability to enter cells. The molecular weight was limited to less than 500 Da [24,39,40,41]. Nitro groups were limited to avoid possible toxicophoric groups [32]. The preparation of the ligands involved adjusting their protonation states and calculating their initial 3D conformations. For each ligand, 30 conformers were generated using the UNITY program within the Sybyl X 2.1 [42] package, which was used for pharmacophoric-based virtual screening using the prepared ligand library.

At the next stage, molecular docking was used to identify compounds that could bind to potential binding sites found in the pharmacophore results. Before the docking simulations, the ADP-ribose and water molecules were removed from the binding site in the model, but the zinc ion and NAD^+^ cofactor were kept. The structure was prepared prior to the MD simulation. The docking was done using the GOLD 5.6 program [43] within a binding site defined around 8 Å from NAD^+^. A genetic algorithm was employed to generate 30 different poses with default parameters. Ligands were ranked by the docking score (GOLD-Score [44]) and ligand affinity (docking score divided by the molecular weight), and the best-ranked ligands were visually inspected and underwent MD simulations to ensure stable interaction pattern [45]. Simulations were conducted as previously described and further elaborated in the Appendix A. Compounds that remained stable within the binding site during the simulations were further considered for acquisition.

### 3.2. Experimental Validation of Compound Inhibition

#### 3.2.1. Chemicals and Biological Reagents

Cell culture medium and supplements were purchased from Hi-Media Laboratories (Delhi, India). Acetylated peptide was synthesized from Aminotech Co. (São Paulo, Brazil). All other reagents were purchased from Sigma Aldrich (St. Louis, MO, USA). The microplate was purchased from Corning Life Sciences (Tewksbury, MA, USA).

#### 3.2.2. *Tc*Sir2 Recombinant Expression

The DNA segment encoding the full-length *Tc*Sir2 (XP_816094) cloned into the plasmid pET24a^(+)^ was previously available in our laboratory. This vector codes for *Tc*Sir2 with a Poly-L-histidine-tag in the C-terminal end. The plasmid was transformed into chemically competent *E. coli* (DE3) ArticExpress cells using heat shock method. Briefly, cells and plasmid were incubated for 30 min on ice, 40 s at 42 °C, followed by another 5 min on ice. Cells containing the plasmid were grown at 30 °C and 150 rpm in Luria Broth medium (LB) containing 30 μg/mL kanamycin until reaching optical density at 590 nm of 0.6. Afterwards, the protein production was induced by adding 0.1 mM isopropyl β-thio-galactopyranoside (IPTG), and cells were further incubated at 12 °C and 200 rpm for 16 h. Cells were pelleted by centrifugation (6000× *g*, 4 °C, 20 min), and the pellet was then resuspended in lysis buffer (25 mM HEPES pH 7.5 with 200 mM NaCl, 5% glycerol, and 5 mM 2-mercaptoethanol). Cell lysis was done by sonication in a Branson Sonifier 250 (Branson instruments, Stanford, CT, USA) using four pulses of 12 s at 30% output potency intercalated with a cool down time (1 min) on an ice bath. Lysate and cell debris were separated by centrifugation (16,000× *g* for 1 h, 4 °C) and the resultant supernatant was added to a pre-equilibrated Ni-NTA resin (Qiagen, Valencia, CA, USA) for 30 min in a cold room. Weakly bound contaminants were removed by washing the resin with the lysis buffer containing 20 mM imidazole. *Tc*Sir2 and chaperone expressed by the ArticExpress (DE3) were eluted with lysis buffer containing 300 mM imidazole. The eluted was then subjected to an ion exchange chromatography with the column MonoQ 5/50 (GE HealthCare) with 50 mM Tris-HCl pH 8 buffer containing a gradient of 0 to 1 M NaCl. Protein concentration was determined by absorption at 280 nm [46] using the extinction coefficient calculated for *Tc*Sir2 [47]. Purified fractions containing proteins, as measured by 280 nm absorption, were applied to a SDS-PAGE [48] to confirm *Tc*Sir2 purity. Purified proteins were transferred to a buffer containing 25 mM Tris-HCl pH 8, 137 mM NaCl, 2.7 mM KCl and 1 mM MgCl_2_ using High Trap Desalting Columns (GE Healthcare, Chicago, IL, USA).

#### 3.2.3. Recombinant *Tc*Sir2 Activity Assay

The peptide substrate (Abz-Gly-Pro-AcetylLys-Ser-Gln-EDDnp, where Abz stands for ortho-aminobenzoic), based on the *Tc*Sir2 specificity [6], was acquired from AminoTech (São Paulo, Brazil). This substrate contains an acetyl-Lys, which is available for trypsin cleavage only when the *Tc*Sir2 deacetylates the lysine side chain. After peptide deacetylation and cleavage, the fluorescent moiety (Abz) and the EDDnp suppressor group (N-[2,4-dinitrophenyl] ethylenediamine) are separated, and fluorescence (420 nm) can occur upon excitation at 320 nm. The *Tc*Sir2 activity assays [6] were performed at 37 °C in a 25 mM Tris-HCl buffer with pH 8 and containing 137 mM NaCl, 2.7 mM KCl, and 1 mM MgCl2. The assays used 0.1 μM purified *Tc*Sir2 enzyme, 10 μM Abz-Gly-Pro-AcetylLys-Ser-Gln-EDDnp peptide, and 25 μM NAD^+^ cofactor. After a 15 min incubation, 12 mM nicotinamide and 0.6 mg/mL trypsin (Sigma Aldrich, Sao Paulo, Brazil, N3376) were added to stop the reaction and cleave the deacetylated peptide (30 min at 37 °C). A control was also prepared without trypsin to detect the baseline fluorescence of the uncleaved peptide. Increasing concentrations of the putative inhibitory compound prepared in DMSO were also tested under the same conditions, with controls to detect the potential effect of DMSO at a maximum concentration of 5% (*v*/*v*) on *Tc*Sir2. Fluorescence was measured using a TECAN-InfinityPro2000 plate reader.

The percentage of inhibition data was calculated using the equation: Inhibition (%) = (1 − v_i_/v_c_) × 100, where v_i_ represents the initial rate in the presence of the putative *Tc*Sir2 inhibitor, and v_c_ is the initial rate of the control assay containing only DMSO. IC_50_ values were calculated based on the percentage of inhibition and concentration of the inhibitor using non-linear regression with least squares on GraphPad Prism (v8.1, GraphPad software, La Jolla, CA, USA). The inhibition mechanism of the most potent compound (CDMS-01) was investigated by determining its effect on the apparent Km and Vmax for NAD^+^ as a substrate, under the same reaction conditions. Km and Vmax in the presence and absence of the inhibitor were obtained by fitting the initial rate and [NAD^+^] into the Michaelis-Menten equation on GraphPad Prism (GraphPad Prism version v8.1, for Windows, GraphPad Software, San Diego, California USA, www.graphpad.com).

#### 3.2.4. Evaluation of In Vitro Trypanocidal Activity with Amastigote Forms

In 96-well plates, cells from the LLCMK2 lineage were seeded at a concentration of 5 × 10^4^ cells/mL (80 μL). The trypomastigote forms of the Tulahuen LacZ strain were added at a concentration of 5 × 10^5^ cells/mL (in a total of 20 μL) and, subsequently incubated for 24 h at 37 °C with a 5% CO_2_ atmosphere. Next, the trypomastigote forms present in the supernatant were washed, leaving only the amastigote forms. Serial dilutions of the compounds were added (1.95 to 200 µM) and the plate was incubated for 72 h at 37 °C with a 5% CO_2_ atmosphere. At the end of this period, the CPRG substrate (chlorophenol red β-D-galactopyranoside, 400 μM in 0.3% Triton X-100, pH 7.4) was added and incubated for additional 6 h. After incubation, the sample’s absorbance values were measured using a spectrophotometer at 570 nm. Benznidazole and dimethyl sulfoxide (DMSO) in the same concentrations as tested compounds were used as the positive and negative controls, respectively.

#### 3.2.5. Mammalian Cell Viability Assay

LLCMK2 cells were maintained in RPMI-1640 medium supplemented with 10% fetal bovine serum (FBS) (Gibco, Carlsbad, CA, USA), penicillin/streptomycin (100 μg.mL^−1^ and 0.1 mg.mL^−1^, respectively, Sigma-Aldrich, St. Louis, Mo, USA) and 2 mM L-glutamine, and treated every second day. At LLCMK2 cells were maintained in high glucose DMEM (Dulbecco’s Eagle’s Medium) medium, supplemented as previously mentioned. All cells were maintained in humidified CO_2_ at 37 °C. Cells were obtained at the third passage, propagated, and used in experiments between the fourth and fifth passages. Cells were routinely checked for contamination with mycoplasma by PCR.

Cytotoxic effects of the compounds were evaluated using MTT (3-(4,5-Dimethylthiazol-2-yl) 2,5-Diphenyl Tetrazolium bromide), which correlates with the cell viability upon treatment. The principle of this method, as described by Mosmann et al. [49], consists in measuring the cellular viability of the enzymatic activity of oxirubutases of living cells [50]. For the test, 1.0 × 10^5^ cells from LLCMK2 lineage were seeded into 96-well microplates in the absence or presence of the compounds or benznidazole (BZN) serially diluted in base 2 (1.95 to 500 μM) and incubated at 37 °C with 5% CO_2_ for 72 h. At the end of the period incubation. 50 μL MTT (Sigma-Aldrich Corp. St. Louis, MO, USA) at the concentration of 2 mg.mL^−1^. After 4 h incubation, 50 μL DMSO per well was added to dissolve formazan blue crystals. The absorbance was determined at 570 nm using Biotek Synergy HT microplate spectrophotometer. The percentage of cell viability was determined from the following formula (40): Cell Viability (%) = (Treatment Absorbance)/(Control Absorbance negative) × 100.

#### 3.2.6. Data Analysis

Statistical comparisons were performed with ordinary or repeated measures of one-way or two-way ANOVA or Friedman test, using respective post hoc tests for multiple comparisons against controls, as recommended by the analysis software and described in the figure legends. IC_50_ values were determined by non-linear fit of dose-response using the equation for sigmoidal dose-response with variable slope.

## 4. Conclusions

The *Tc*Sir2 inhibitor CDMS-01, which is active in vitro against intracellular amastigotes of *T. cruzi*, is a new antitrypanosomal hit compound. For this reason, future studies may apply molecular simplification to increase activity and selectivity. CDMS-01 is a good starting point for the development of new drugs for the Chagas disease. Despite the low and medium micromolar activity against *Tc*Sir2 of our hits, they represent novel scaffold, and their derivatives may be tested to improve on-target activity.

## Figures and Tables

**Figure 1 pharmaceuticals-16-00428-f001:**
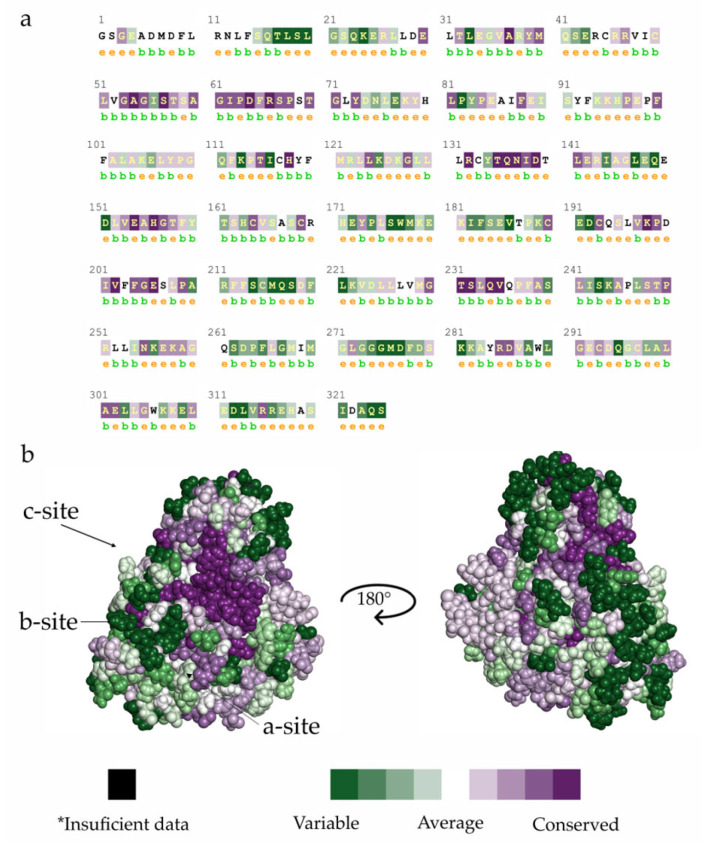
Conservation analysis of *T. cruzi* (*Tc*Sir2) sirtuin 2 compared to human (hSir2) and *L. infantum* (*Li*Sir2). (**a**) The sequence of amino acids (*Tc*Sir2). (**b**) Conserved and not conserved regions of *Tc*Sir2. e—An exposed residue according to the neural-network algorithm; b—a buried residue according to the neural-network algorithm; *—Insufficient data—the calculation for this site was performed on less than 10% of the sequences.

**Figure 2 pharmaceuticals-16-00428-f002:**
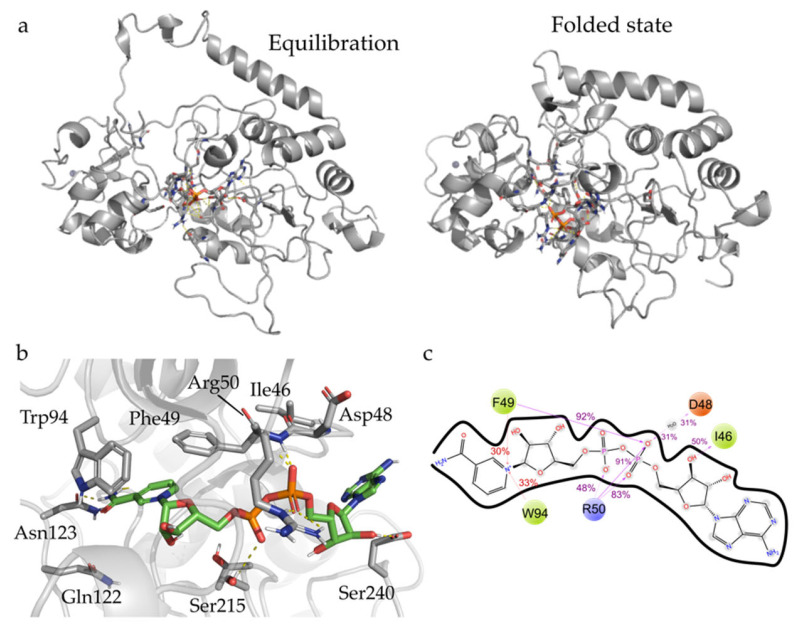
Molecular dynamics analyses. (**a**) Representative snapshots of the simulation showing the differences between the initial model in equilibration and representative frame after MD, highlighting the convergence of the N-terminal region towards the protein’s core; (**b**) cofactor sites and interactions with *Tc*Sir2 and frequency with those residues’ interactions with NAD^+^ along the simulation (**c**), interaction frequency is represented by numbers over the 2D representation.

**Figure 3 pharmaceuticals-16-00428-f003:**
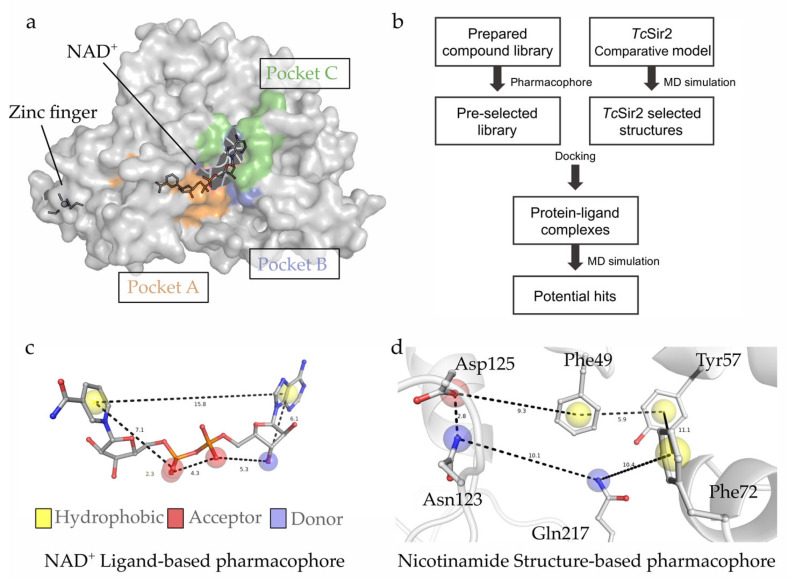
Development of the pharmacophore model. (**a**) *Tc*Sir2 after the comparative modelling, molecular dynamics simulations, and cofactor binding site comprises four regions: zinc finger region, A site (orange), B site (purple), C site (green), and pharmacophore model. (**b**) Workflow for a constructed model to pharmacophore model development. (**c**) Pharmacophore 1 (based on A, B, and C sites) and (**d**) pharmacophore 2 (based only C sites).

**Figure 4 pharmaceuticals-16-00428-f004:**
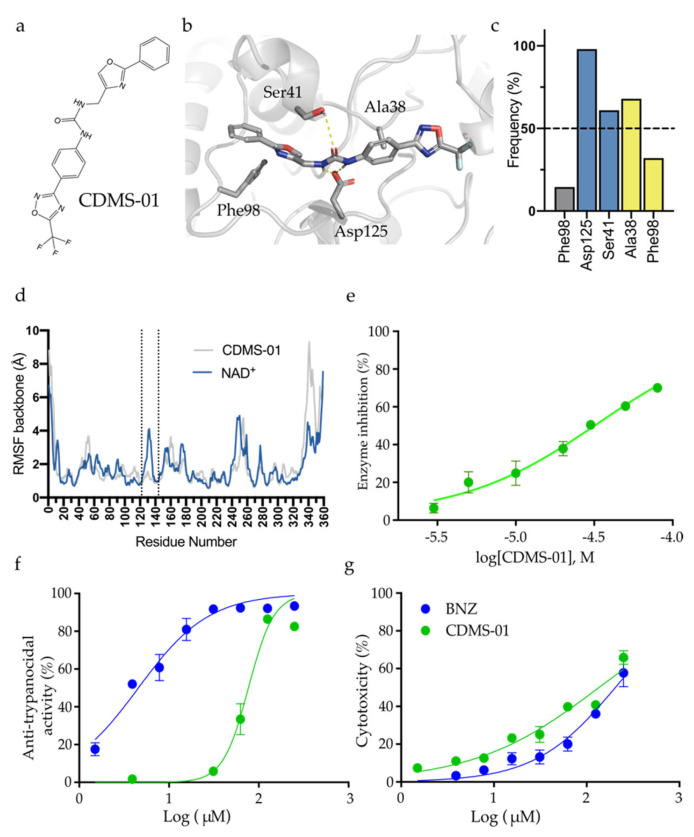
CDMS-01 modelling analyses and experimental validation. (**a**) 2D structure CMDS-01 and Benznidazole, (**b**) CDMS-01 3D structure presents interactions responsible for stability in *Tc*Sir2 (**c**) frequency contacts between CDMS-01 with *Tc*Sir2, and (**d**) residues’ fluctuation of the NAD^+^ and CMDS-01 in *Tc*Sir2, (**e**) dose-response curves of CDMS *Tc*Sir2 inhibition (IC_50_), (**f**) trypanocidal activity on amastigote (IC_50_ in comparison with benznidazole as positive control), and (**g**) cytotoxicity against LLCMk2 cell line (EC_50_ in comparison with benznidazole as positive control).

**Table 1 pharmaceuticals-16-00428-t001:** IC_50_ (*Tc*Sir2), IC_50_ (on *T. cruzi* amastigote), and EC_50_ (LLMCK2) analysis for the compounds CDMS-02 to CDMS-06; SI stands for selectivity index (EC_50_/IC_50(ama)_). **ND** Enzyme values were measured but did not converge IC_50_ significantly. ND* Cell assay values were measured but did not converge IC_50_ significantly.

Compound	IC_50_ (µM)	IC_50_ *T. cruzi* ama (µM)	EC_50_ Cell Toxicity (µM)	SI_ama_
**CDMS-01**	39.98 ± 0.05	76.68 ± 0.02	135.91 ± 0.50	1.77
**CDMS-02**	240.21 ± 1.25	183.91 ± 0.01	100.13 ± 0.10	0.54
**CDMS-03**	482.21 ± 1.35	163.17 ± 0.02	245.91 ± 0.13	1.5
**CDMS-04**	**ND**	15.16 ± 0.03	62.35 ± 0.15	4.11
**CDMS-05**	**ND**	177.17 ± 0.03	239.97 ± 0.31	1.35
**CDMS-06**	332.13 ± 0.55	169.40 ± 0.02	226.83 ± 0.56	1.33
**Nicotinamide**	987.4 ± 0.64	ND*	ND*	ND*
**BZN**	9.33 ± 0.01	4.55 ± 0.03	204.05 ± 0.81	44.8

## Data Availability

Appendix A is available at Correspondence and requests for materials should be addressed to G.M.F. and T.K.

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
