# Peer review of "Trypanosoma cruzi Sirtuin 2 as a Relevant Druggable Target: New Inhibitors Developed by Computer-Aided Drug Design"

_pharmaceuticals, 2023, doi:10.3390/ph16030428_

Round 1

Reviewer 1 Report

The ms by Ferreira et al. describes a VS approach to identify new T. cruzi Sirt2 inhibitors able to counteract the protozoan infection in cells. It shows a shows a lenghty and detailed description of he computational methods used to identify new cpds as TcSirt2 inhibitors, followed by a quick investigation of their effects against the enzyme and in T. cruzi infected cells. Out of 6 cpds identified by the screening, only 1 displayed effective inhibition of the target, and moderate activity in infected cells joined to some toxicity. The potency as well as the selectivity index of the selected compd is low thus this should be retained a preliminary note. In the main text there are a lot of mistakes with the english language, and sometimes certain phrases (for instance, at pag 12 line 480, pag 14 lines 513-515, pag 14 lines 534-538, and others) are incomprehensible at all and should be totally rewritten. The discussion around the chemical structure of the hit talks about fluoro-oxazole but it shows a trifluoromethyl group...many times the singular is confused with the plural and vice versa. More importantly, the SI is too low to allow in vivo study, contrarily to what written by the authors. Thus, the whole ms need a accurate and precise revision before being considered for publication

  FileEditViewFormat Paragraph                      

Author Response

REVIEWER #1

The ms by Ferreira et al. describes a VS approach to identify new T. cruzi Sirt2 inhibitors able to counteract the protozoan infection in cells. It shows a show a lengthy and detailed description of the computational methods used to identify new cpds as TcSirt2 inhibitors, followed by a quick investigation of their effects against the enzyme and in T. cruzi infected cells. Out of 6 cpds identified by the screening, only 1 displayed effective inhibition of the target, and moderate activity in infected cells joined to some toxicity. The potency as well as the selectivity index of the selected compd is low thus this should be retained a preliminary note. In the main text there are a lot of mistakes with the English language, and sometimes certain phrases (for instance, at pag 12 line 480, pag 14 lines 513-515, pag 14 lines 534-538, and others) are incomprehensible at all and should be totally rewritten. The discussion around the chemical structure of the hit talks about fluoro-oxazole but it shows a trifluoromethyl group...many times the singular is confused with the plural and vice versa. More importantly, the SI is too low to allow in vivo study, contrarily to what written by the authors. Thus, the whole ms need a accurate and precise revision before being considered for publication

Authors response: Thanks for the opportunity of further revise our work and we tried in the following sessions to address your comments. In addition, we are also submitting a manuscript version where all modified parts are highlighted in yellow.

Reviewer 2 Report

In the manuscript titled “Trypanosoma cruzi Sirtuin 2 as a relevant druggable  target: new inhibitors developed by computer-aided drug design” it is reported  the modelling of TcSir2, comparing it with the known structures of human and Leishmania infantum  Silent Information Regulator 2. In addition the authors have studied by virtual screening a library of commercial available compounds deriving from ZINC15 database. After a physicochemical filter and molecular docking calculation, the authors have selected six molecules to be bio-assayed in vitro. The most promising molecule was subjected to molecular dynamic simulation.

The paper is well organized. The introduction reports the state of art in the knowledge of Sir2 and its potential druggable target. The obtained results are encouraging.

Minor comments are:

1. at row 141 please put superscript the charge of ions

2. at row 161 put subscript “2” in O2 and H2O

3. In Figure 2, please increase the resolution of images (a) and (c).  

In conclusion, I recommend to accept this paper for publication after correct minor issues.

Author Response

REVIEWER #2

In the manuscript titled “Trypanosoma cruzi Sirtuin 2 as a relevant druggable target: new inhibitors developed by computer-aided drug design” it is reported the modelling of TcSir2, comparing it with the known structures of human and Leishmania infantum Silent Information Regulator 2. In addition, the authors have studied by virtual screening a library of commercial available compounds deriving from ZINC15 database. After a physicochemical filter and molecular docking calculation, the authors have selected six molecules to be bio-assayed in vitro. The most promising molecule was subjected to molecular dynamic simulation.

The paper is well organized. The introduction reports the state of art in the knowledge of Sir2 and its potential druggable target. The obtained results are encouraging.

In conclusion, I recommend accepting this paper for publication after correcting minor issues.

Authors response: Thanks for the opportunity of further revise our work and we tried in the following sessions to address your comments. In addition, we are also submitting a manuscript version where all modified parts are highlighted in yellow.

Minor comments are:

Q1. at row 141 please put superscript the charge of ions

Authors response: We are also submitting a manuscript version where all modified parts are highlighted in yellow.

QQ2. at row 161 put subscript “2” in O2 and H2O

Authors response: We are also submitting a manuscript version where all modified parts are highlighted in yellow.

Q3. In Figure 2, please increase the resolution of images (a) and (c). 

Authors response: We are also submitting a manuscript version where all modified on the figures are done.

Reviewer 3 Report

In this manuscript, the authors propose the use Sir2 enzyme as a drug target to interfere with the parasites’ (Trypanosoma cruzi) cell cycle. The molecular dynamic and molecular docking were applied to investigate potential inhibitors. For the six best-scoring compounds biological assay was performed. In my opinion, the work is well thought out and planned. The methods used are appropriate. The results were explained understandably. I have no significant substantive comments. I think that the obtained results can be a good background for further research on new compounds against Chagas disease. However, a few minor remarks:

-figure 2c is very low resolution, Figures 2b and 2b could be more readable.

- line 597, ref 8 -???  something is wrong, (this paper has only 5 authors)

Author Response

REVIEWER #3

In this manuscript, the authors propose the use Sir2 enzyme as a drug target to interfere with the parasites’ (Trypanosoma cruzi) cell cycle. The molecular dynamic and molecular docking were applied to investigate potential inhibitors. For the six best-scoring compounds biological assay was performed. In my opinion, the work is well thought out and planned. The methods used are appropriate. The results were explained understandably. I have no significant substantive comments. I think that the obtained results can be a good background for further research on new compounds against Chagas disease. However, a few minor remarks:

Authors response: Thanks for the opportunity of further revise our work and we tried in the following sessions to address your comments. In addition, we are also submitting a manuscript version where all modified parts are highlighted in yellow.

Q1. figure 2c is very low resolution, Figures 2b and 2b could be more readable.

Authors response: We are also submitting a manuscript version where all modified on the figures are done

Q2. line 597, ref 8 -???  something is wrong, (this paper has only 5 authors)

Authors response: Thanks for the opportunity of further revise our work and we tried in the following sessions to address your comments. In addition, we are also submitting a manuscript version where all modified parts are highlighted in yellow.

Reviewer 4 Report

This work done by Ferreira et al developed a series of new inhibitors to TcSir2 by computer-aided drug design and CDMS-01 showed to be a potential leading compound. Though its potent is weak but it showed some selectivity and opened up a new avenue for innovative drug development for Trypanosoma cruzi infection. I listed a few suggestions to further improve the quality of this work:

“Compounds with net charge beyond -2 and 2 and larger than 500 Da were removed from the initial library aiming to enrich for compounds with better cellular permeability”: Are the net charge and molecular weight the only criteria to predict the cellular permeability? if not, what are the other determinants for their cellular permeability? And how does CDMS-01 fulfill them?

“The ADP-ribose and structural water molecules were removed from the model’s binding site prior to the docking simulations”: have you done the competition experiment between CDMS-01 and ADP-ribose regarding binding to the putative binding pocket?

“TcSir2 was modeled using the human Sir2 (PDB: 3ZGV) structure as template”: is 46% similarity between TcSir2 and human Sir2 sufficient to model the TcSir2 on top of the Sir2 structure. 

Is it a good idea to use alpha-fold to predict the structure of TcSir2 and do modeling direct on the TcSir2?

what the PAN-inhibitors mean and why should avoid them?

“Six compounds fitted these criteria (CDMS-05 and CDMS-06 (pharmacophore model 1 – Fig. 3c) and CDMS-01 to CDMS-04 (pharmacophore model 2 – Fig. 3d) and were selected for experimental validation”: water solubility and cell permeability of these compounds in the targeted cell lines (or only show CDMS-01)?

“CDMS-01 is an innovative prototype acting on a new druggable target that can be a starting point to search for new inhibitors that have better activity and selectivity”: discuss the possible directions for the optimization of the activity and selectivity.

“some compounds even with moderate SI values, are approved by the FDA, for example, Amphotericin B (SIama = 0.15) and Meglumine antimoniate (SIama = 2.46)”: list the IC50 of the two FDA-approved drugs and compared it with your compounds. The SI of Amphotericin B is < 1, does it mean this drug presents no selectivity at all and how does it explain the therapeutic efficacy? 

The mechanism of CDMS-04 selectively against amastigote. 

Author Response

REVIEWER #4 This work done by Ferreira et al developed a series of new inhibitors to TcSir2 by computer-aided drug design and CDMS-01 showed to be a potential leading compound. Though its potent is weak but it showed some selectivity and opened a new avenue for innovative drug development for Trypanosoma cruzi infection. I listed a few suggestions to further improve the quality of this work:   Authors response: Thanks for the opportunity of further revise our work and we tried in the following sessions to address your comments. There are a point-by-point list of our major conclusions and some representative citations from the paper, where extensively modifications were done. In addition, we are also submitting a manuscript version where all modified parts are highlighted in yellow.     Q1. “Compounds with net charge beyond -2 and 2 and larger than 500 Da were removed from the initial library aiming to enrich for compounds with better cellular permeability”: Are the net charge and molecular weight the only criteria to predict the cellular permeability? if not, what are the other determinants for their cellular permeability? And how does CDMS-01 fulfill them?   Authors response: Thanks for the opportunity of further revise our work and we tried in the following sessions to address your comments. There are a point-by-point list of our major conclusions and some representative citations from the paper, where extensively modifications were done. We used Filter by OpenEye to run and selected keep these rules (Lipinski, net charger (-2/+2) and molecular weight). In addition, we are also submitting a manuscript version where all modified parts are highlighted in yellow.       Q2. “The ADP-ribose and structural water molecules were removed from the model’s binding site prior to the docking simulations”: have you done the competition experiment between CDMS-01 and ADP-ribose regarding binding to the putative binding pocket?   Authors response: Thanks for the opportunity of further revise our work and we tried in the following sessions to address your comments. Unfortunately, not, because our assay is a “indirect” assay, we mensurated the activity after the mechanism of the Sirtuin 2, when we added the stop solution and we mensurated the trypsin active clivate the domain Pro-Lys on the substrate. While the Pro-Lys(Ac) do not clivate by trypsin.     Q3. “TcSir2 was modeled using the human Sir2 (PDB: 3ZGV) structure as template”: is 46% similarity between TcSir2 and human Sir2 sufficient to model the TcSir2 on top of the Sir2 structure.    Authors response: Thanks for the opportunity of further revise our work and we tried in the following sessions to address your comments. We used the hSir2 to find similar and different regions, while try search a selective compound to tcSir2 than hSir2.   Q4. Is it a good idea to use alpha-fold to predict the structure of TcSir2 and do modeling direct on the TcSir2?   Authors response: Thanks for the opportunity of further revise our work and we tried in the following sessions to address your comments. However, we ran a blastP on NCBI dependences and found these results:   Organism Similarity with TcSir2 % PDB code Crystal resolution R-value free R-value work Ligands NADH Leishmania infantum 58.075 5OL0 1.99 0.22 0.16 Peptide linking No Homo sapiens 46.186 5D7O 1.63 0.19 0.17 No Yes Homo sapiens 46.186 4L3O 2.52 0.26 0.21 macrocyclic peptide No Homo sapiens 46.186 4X3O 1.5 0.15 0.12 myristoyl peptide No Homo sapiens 46.186 1J8F 1.7 0.26 0.23 No No Homo sapiens 46.186 6QCN 2.23 0.23 0.20 quercetin Yes Homo sapiens 46.186 5YQL 1.6 0.18 0.15 A2I (selective) No Homo sapiens 46.186 3ZGV 2.27 0.18 0.14 No Yes Homo sapiens 46.186 5MAR 1.89 0.19 0.17 1,2,4-Oxadiazole Yes Homo sapiens 46.186 4RMG 1.88 0.24 0.20 SirReal2 Yes Homo sapiens 45.868 4Y6L 1.6 0.23 0.20 H3K9myr No Homo sapiens 45.798 4R8M 2.1 0.27 0.22 BHJH-TM1 No Homo sapiens 45.763 3ZGO 1.63 0.19 0.16 No No   Meanwhile, we expected that the template pdb: 3ZGV is better as alphafold models (https://alphafold.ebi.ac.uk/entry/Q8IXJ6). Further, the alignment with models keeps the same folding than TcSir2 (https://www.ebi.ac.uk/pdbe/pdbe-kb/proteins/Q8IXJ6). Bellow the alignment of backbone with all the Sirtuin 2 structures from PDB against our model (TcSir2 model highlighted in gray; Figure Q4).     Figure Q4. Backbone alignment with all the Sirtuin 2 structures from PDB against our model (TcSir2 model highlighted in gray) (manuscript enclosed).     Q5. what the PAN-inhibitors mean and why should avoid them?   Authors response: Thanks for the opportunity of further revise our work we tried in the following sessions to address your comments. We noticed that because of various biological pathways, the sirtuin enzyme family members SIRT1, SIRT2, and SIRT3 play different pathways in trypanossomatides concerning expression, based on the context and experimental conditions. Hence, an interesting question is whether inhibiting one of them or inhibiting all of them would be better for treating Chagas disease, while Sirtuin 2 is more expressed on amastigote form, the form concerning the chronic phase disease. Pharmacologically, this is difficult to address, due in part to potential off-target effects of different compounds. Compounds with almost identical properties but differing in SIRT1–3 selectivity will be useful for addressing this question. In addition, we are also submitting a manuscript version where all modified parts are highlighted in yellow.     Q6. “Six compounds fitted these criteria (CDMS-05 and CDMS-06 (pharmacophore model 1 – Fig. 3c) and CDMS-01 to CDMS-04 (pharmacophore model 2 – Fig. 3d) and were selected for experimental validation”: water solubility and cell permeability of these compounds in the targeted cell lines (or only show CDMS-01)?   Authors response: Thanks for the opportunity of further revise our work we tried in the following sessions to address your comments. Exactly, CDMS-01 is only show the cell permeability and active in cell assay, as presented in table 1 (line 492-497).     Q7. “CDMS-01 is an innovative prototype acting on a new druggable target that can be a starting point to search for new inhibitors that have better activity and selectivity”: discuss the possible directions for the optimization of the activity and selectivity.   Authors response: Thanks for the opportunity of further revise our work we tried in the following sessions to address your comments. We discuss that in the lines (526-530 / 539-544), while the toxicity and lower SIama than BZN (Table 1). Howerver, a molecular simplification keeps while the bulkier substituents could reduce the toxicity and increase the SI.                                                                                                                                                              Q8. “some compounds even with moderate SI values, are approved by the FDA, for example, Amphotericin B (SIama = 0.15) and Meglumine antimoniate (SIama = 2.46)”: list the IC50 of the two FDA-approved drugs and compared it with your compounds. The SI of Amphotericin B is < 1, does it mean this drug presents no selectivity at all and how does it explain the therapeutic efficacy?    Authors response: Thanks for the opportunity of further revise our work we tried in the following sessions to address your comments. In addition, we are also submitting a manuscript version where all modified parts are highlighted in yellow (Line 525-527)     Q9. The mechanism of CDMS-04 selectively against amastigote.    Authors response: Thanks for the opportunity of further revise our work we tried in the following sessions to address your comments. We noticed that, but the values for selective is far from BZN, values higher than 40 are considered selective cpds against trypanosotides. While SIR2 expression level on amastigotes is higher in comparison to epimastigotes

Round 2

Reviewer 1 Report

The reply by the authors through a point-by point rebuttal letter is missing. Nothing of the issues raised from the first review round has been accomplished. My opinion is that this manuscript does not deserve pblication on Pharmaceuticals

Author Response

1- Revise the English language and style: I have made text edits directly on the pdf to help you with this task (see attached file).

Authors response: Thank you for the opportunity to revise our manuscript. We have taken into consideration the comments and suggestions provided by the reviewers and have made the necessary changes in the revised version. All modifications are highlighted in yellow for easy reference. We appreciate your time and consideration, and hope that this updated version meets the standards for publication.

2- Abstract: a suggestion to amend the abstract appears below.
"Trypanosoma cruzi, the etiological agent of Chagas disease relies on finely coordinated epigenetics regulation during the transition between hosts. Herein we targeted the Silent Information Regulator 2 (Sir2) enzyme, a NAD+-dependent class III histone deacetylase, to interfere with the parasites’ cell cycle. A combination of molecular modelling with on-target experimental validation was used to discover new inhibitors from commercially available compound libraries. We selected six inhibitors from the virtual screening, which were validated on recombinant Sir2 enzyme. The most potent inhibitor (CDMS-01, IC50 = 40 μM) was chosen as a potential lead compound."

Authors response: Thanks for the opportunity of further revise our work and we tried in the following sessions to address your comments. In addition, we are also submitting a manuscript version where all modified parts are highlighted in yellow.

3- Line 533-536 (“Furthermore...nitrile”): even though this hypothesis could stand to explain an in vivo effect of the compounds with such low in vitro activity, it is not clear why it is mentioned here. In the absence of such in vivo data, this needs to be explained or the sentence deleted.

Authors response: Thank you for the opportunity to revise our manuscript. We have taken into consideration the comments and suggestions provided by the reviewers and have made the necessary changes in the revised version. All modifications are highlighted in yellow for easy reference.

4- Line 552-554 (“While the SI values showed no selectivity against TcSir2...target”): this sentence is unclear. The SI values reported in the manuscript refer to the selectivity between T. cruzi amastigotes and the mammalian cell line LLCMK2, not towards TcSir2 vs other isoforms. Please revise.

Authors response: Thank you for the opportunity to revise our manuscript. We have taken into consideration the comments and suggestions provided by the reviewers and have made the necessary changes in the revised version. It was reported by Schenckman et al 2015 (ref 7 on the manuscript) there are two sirtuins in T. cruzi, TcSir2rp1 and TcSir2rp3. They found that TcSir2rp1 is localized in the cytosol and TcSir2rp3 in the mitochondrion. TcSir2rp1 overexpression acts to impair parasite growth and differentiation. All modifications are highlighted in yellow for easy reference. We appreciate your time and consideration, and hope that this updated version meets the standards for publication.

5- Line 559 (“future studies may apply molecular simplification to increase activity and selectivity”): this sentence is unclear. What does “molecular simplification” mean? Do you mean: “Future structure-activity relationships studies may be performed to increase activity and selectivity”?

Authors response: We would like to express our gratitude for the opportunity to revise our manuscript. We have carefully considered the comments and suggestions from the reviewers and have made the necessary modifications in the revised version. We have incorporated one of the reviewer's suggestions into the text. All the changes have been marked in yellow for easy reference. We deeply appreciate your time and attention and hope that this revised version meets the required standards for publication.

Reviewer 4 Report

accept in present form

Author Response

We sincerely appreciate your feedback and the comments.